# Is Computer-Assisted Tissue Image Analysis the Future in Minimally Invasive Surgery? A Review on the Current Status of Its Applications

**DOI:** 10.3390/jcm10245770

**Published:** 2021-12-09

**Authors:** Vasilios Tanos, Marios Neofytou, Ahmed Samy Abdulhady Soliman, Panayiotis Tanos, Constantinos S. Pattichis

**Affiliations:** 1Department of Obstetrics and Gynecology, Aretaeio Hospital, 2024 Nicosia, Cyprus; 2St. Georges’ Medical School, University of Nicosia, 2408 Nicosia, Cyprus; soliman.a@live.sgul.ac.cy; 3Biomedical Engineering Research Center, Department of Computer Science, University of Cyprus, 1678 Nicosia, Cyprus; mneoph@ucy.ac.cy (M.N.); pattichi@ucy.ac.cy (C.S.P.); 4Medical School, University of Aberdeen, Foresterhill Rd., Aberdeen AB25 2ZD, UK; p.tanos.17@abdn.ac.uk

**Keywords:** tissue image analysis, tissue texture image analysis, optical biopsies, computer-aided diagnosis

## Abstract

Purpose: Computer-assisted tissue image analysis (CATIA) enables an optical biopsy of human tissue during minimally invasive surgery and endoscopy. Thus far, it has been implemented in gastrointestinal, endometrial, and dermatologic examinations that use computational analysis and image texture feature systems. We review and evaluate the impact of in vivo optical biopsies performed by tissue image analysis on the surgeon’s diagnostic ability and sampling precision and investigate how operation complications could be minimized. Methods: We performed a literature search in PubMed, IEEE, Xplore, Elsevier, and Google Scholar, which yielded 28 relevant articles. Our literature review summarizes the available data on CATIA of human tissues and explores the possibilities of computer-assisted early disease diagnoses, including cancer. Results: Hysteroscopic image texture analysis of the endometrium successfully distinguished benign from malignant conditions up to 91% of the time. In dermatologic studies, the accuracy of distinguishing nevi melanoma from benign disease fluctuated from 73% to 81%. Skin biopsies of basal cell carcinoma and melanoma exhibited an accuracy of 92.4%, sensitivity of 99.1%, and specificity of 93.3% and distinguished nonmelanoma and normal lesions from benign precancerous lesions with 91.9% and 82.8% accuracy, respectively. Gastrointestinal and endometrial examinations are still at the experimental phase. Conclusions: CATIA is a promising application for distinguishing normal from abnormal tissues during endoscopic procedures and minimally invasive surgeries. However, the efficacy of computer-assisted diagnostics in distinguishing benign from malignant states is still not well documented. Prospective and randomized studies are needed before CATIA is implemented in clinical practice.

## 1. Introduction

Clinicians and computer scientists have been exploring the possibilities of using computer-assisted tissue image analysis (CATIA) in distinguishing normal from abnormal tissues. The fast development and application of endoscopy for diagnosis and treatment in daily medical practice, and the use of high-definition video recording facilitate research on CATIA [1]. Processing evaluation is based on a manual or automated image interpretation that filters artefacts from a database of images. We reviewed existing studies to evaluate the potential of performing optical biopsies of human tissues and the reliability of the results, evaluated the gained experience from CATIA on various tissues, and explored the possibilities of implementing tissue image texture analysis in daily medical practice within the context of computer-assisted diagnostics (CAD) during minimally invasive surgery. In the studies reviewed, tissue image processing techniques focus on three aspects. Tissue image features, colour-spectrum characterization and filtering as well as algorithm and statistical evaluation during or after a patient’s endoscopic procedure (Table 1).

Examining suspicious tissue without an invasive procedure by in vivo optical biopsy with tissue image analysis has several advantages: preventing vascular and tissue injury, haemorrhage, haematoma, spread of malignant cells, and risk for infection and scarring. In addition, CATIA allows for comparing suspicious tissue to its neighbouring healthy regions. Thus far, final diagnosis and treatment follows histopathologic examination. Nevertheless, tissue image analysis may guide the physician during biopsy sampling by providing a high-risk or low-risk tissue malignancy score. CATIA could serve as a ‘second opinion’, augmenting the physician’s decision on the biopsy sampling location and allowing a preliminary tissue image result. In the suspicion of malignancy, the histopathological examination can be prioritised. Moreover, tissue image analysis could decrease the risk for error, especially in difficult and suspicious cases with extensive visual tissue variability [2]. 

Tissue images during minimally invasive surgery and endoscopic images during colonoscopy and gastroscopy for CATIA could be manually isolated and evaluated when an abnormality or lesion is suspected. An automated system could also be used to define frames of normal and abnormal characteristics in endoscopic segments with different visual appearance [3]. An expert might easily choose the frames that need further processing, though the inexperienced practitioner might find this more difficult. Specific tissue segments could be isolated from a video, and groups of frames with suspicious pathologic features could be visualized and selected for CATIA. Clustering and classification techniques facilitate the selection of automated images and allows surveillance of defined targets [1]. Skin, gastrointestinal tract [GIT], larynx, and endometrial [4,5] tissues were analysed for malignancy by optical biopsies. 

The purpose of the study was to provide a review of computer-assisted tissue image analysis studies during minimally invasive surgery and endoscopy. In addition, we review and evaluate the impact of in vivo optical biopsies performed by tissue image analysis on the surgeon’s diagnostic ability and sampling precision.

## 2. Materials and Methods

We reviewed the scientific literature for articles in which CATIA was used to distinguish normal from abnormal human tissues, screen for high-risk cancer cases, and assist in diagnosis and treatment. Using the keywords ‘computer-assisted tissue-image analysis’, ‘computer-aided-diagnosis-systems’, ‘tissue-texture-analysis’, ‘tissue texture image analysis’, ‘endoscopy’ and by filtering with words such as ‘computer-assisted-diagnosis’ and ‘tissue-texture-analysis’, we searched the scientific databases PubMed, IEEE, Xplore, Elsevier, and Google Scholar for relevant articles from 2010 to 2020. Last day of access was the 16th of July 2021. Our search further specifically isolated the articles in the disciplines of dermatology, gastroenterology, otorhinolaryngology (ENT), and gynaecology. No automation tools were used in the process. 

Articles which did not analyse tissue, reported no tissue texture parameters, as well as any article dealing with numerical measurements or algorithms were excluded from our study. Articles dealing with computing and virtualization were disregarded from our review as well as more complicated, machine learning techniques using colour spectra and filters. However, papers that used tissue texture analysis and augmented their diagnostic methodology using colour spectra or other technologies were included in this review. The isolated articles included digital analysis form, computation analysis, texture image analysis, or any other form of digital CAD specifically on tissue. Each of these forms of analysis use different tools and methods of scrutiny and, therefore, an individualized approach was performed to isolate the relevant articles as described in Figure 1. A total of 28 relevant articles were selected on the basis of CAD methods used and tissue examined, the data are summarized in Table 2, Table 3 and Table 4 in order to be assessed and compared. This includes references, CAD method, sample-size, results, conclusions, and critical comments. 

In most studies, the numerical and quantifiable values produced by tissue texture analysis and interpreted by algorithms came from endoscopic observation of a suspected abnormal tissue region and were correlated with the histopathologic results of a tissue biopsy under light microscopy. Most studies used the algorithms derived from correlating CATIA results with histopathologic criteria for malignancy in individual tissues and relied on computer-assisted diagnostic methods and tissue analysis to distinguish pre-malignant from malignant tissues based on repetition, number of patients, and abnormal tissue variability.

All images used in the studies were of human tissue. The images were captured by photography in dermatologic and histopathologic cases, by endoscopy with an external and capsule camera in the gastrointestinal cases, and with a 3-chip camera during hysteroscopy and laparoscopy. All studies focused on identifying neoplastic disease by using CATIA to compare tissue image optical criteria with histopathologic results. 

A few studies compared normal with abnormal regions of interest captured from the same images and compared to histopathologic findings. Pre-processing procedures were used for reducing image noise, such as γ-correction algorithm (Digital image processing algorithm that compensates for the nonlinear effect of signal transfer between electrical and optical devices) and liner image stretching (Point operation in digital image processing to improve an image by stretching the range of intensity values to a desired range of values). Furthermore, image partitioning (Image segmentation to smaller image parts), was used to select the Regions of Interest (ROIs). Image discriminating texture features (Method to extract relevant information from an image), used in all studies, tried to isolate the best single algorithm or combination of algorithms to distinguish benign from malignant tissue. Manual image segmentation was performed in most studies. Automated segmentation was reported in gastrointestinal videos. Texture and colour discrimination characteristics, multistage illumination (different image scale sizes), correction algorithms (Digital image processing luminance correction to a range of values), histogram equalization (Method that normalises the histogram values to a specific range of values), and support vector machine (SVM) algorithms (Supervised machine learning algorithm that solves two group classification problems) were the main tools used for image analysis.

## 3. Results

### 3.1. Assessing Dermatological Abnormalities 

Skin diseases evaluated by using CATIA focus on distinguishing malignant from benign conditions. One study on basal cell carcinoma (BCC), two on squamous cell carcinoma (SCC), three on melanoma, one on vitiligo, and one on skin lesions were reported (Table 2). CATIA distinguished normal from benign and malignant tissues with an accuracy of 90% and diagnostic accuracy of 94%, among 80 skin samples of BCC and SCC. In another study of tissue image analysis, the accuracy of distinguishing nevi melanoma from benign disease was from 73% to 81% [4,5,6]. Raman spectroscopy and principal component algorithm were used on skin images and distinguished normal from BCC and melanoma with an accuracy of 92.4%, sensitivity of 99.1%, and specificity of 93.3%. In another study focused on texture image analysis of melanoma versus normal and benign precancerous lesions, discrimination accuracy was 91.9% and 82.8%, respectively [7]. 

The principal component analysis (PCA) algorithm, used in microscopy images from histopathologic sections to discriminate 145 different spectra of skin biopsies between BCC and melanoma, exhibited accuracy of 92.4%, sensitivity of 99.1%, and specificity of 93.3% [6,8]. A combination of PCA-hLDA and Raman spectroscopy, followed by tissue linear discriminant analysis classification models, demonstrated accuracy of 86%, sensitivity of 100%, and specificity of 66% in differentiating oral cavity SCC of the tongue [17]. Computer-assisted tissue texture analysis (CATTA) of 80 microscopy images of 200 × 200 pixels recognition against BCC provided 89.92% for allocation accuracy and 93.75% for diagnosis accuracy in locating the region of interest (ROI) for the lesion and discriminating texture features [5]. PCA’s use was further recognized in BCC and melanoma discrimination from normal skin biopsies in vitro. PCA and elucidation distance classified samples according to histopathologic features with an accuracy of 92.4%, sensitivity of 99.1%, and specificity of 93.3%, by using Raman spectroscopy coupled with a fiber optic Raman probe [6,8].

Similarly, Silveria FL et al., 2015, successfully discriminated non melanoma skin lesions from non tumour human skin tissues in vivo before surgery by using Raman spectroscopy and multivariate statistics [7]. PCA/DA and PLS/DA, based on mathematical metric space distances, accurately distinguished non melanoma from normal and benign precancerous lesions with 91.9% and 82.8% accuracy, respectively. High-frequency ultrasound and ultrasound elastography (USE) were used to distinguish benign from malignant skin lesions, and histopathologic evaluation for malignancy was correlated with a ratio of compressibility. In characterizing these lesions as malignant, a diagnostic value of 3.0 to 3.9 resulted in 100% sensitivity and specificity [18]. Computerized algorithms were also used in the primary diagnosis and recurrence management of vitiligo. Vitiligo patients’ primary treatment efficacy was assessed by the digital method (MPR-CDIAS) and proved to perform objective analysis of repigmentation or depigmentation in vitiligo skin lesions in response to treatment. 

### 3.2. Assessing Gastrointestinal Track Abnormalities 

Most CATIA studies we reviewed concern research on GIT endoscopy (Table 3). Tissue image analysis from a wireless endoscopy capsule of 14 patients’ small bowels, reported high specificity and sensitivity in distinguishing malignant from benign tumours [4]. Oesophageal metaplasia was seen in the images of 34 patients undergoing esophagoscopy using 3D optical coherence tomography (OCT) and the detection rate of metaplasia increased by 72% [4]. Chromo endoscopy images using CAD tools compared to standard colonoscopy images and histopathologic biopsies improved the detection rate of celiac disease diagnosis by 10%, the detection of dysplastic cells among inflammatory bowel disease by 7%, and of dysplasia in polyp and ulcerative colitis by 9.3% and 1.3%, respectively [9]. Celiac disease diagnosis was achieved by using gastroscopy and CATIA in identifying mucosa alterations in 2835 duodenum images from 290 children [9]. Using 350 oesophageal and 129 gastric mucosa images captured by fiber optic telescope, an algorithm system application diminished the honeycomb effect on final images, thus improving the clarity and diagnosis [19]. 

### 3.3. Assessing Endometrial Hyperplasia and Cancer and Ovarian Malignancies 

On a previous study, we identified the optimal conditions of image capturing during hysteroscopy and laparoscopy [10] and the ability of a CAD system to distinguish normal from abnormal epithelia by using animal models and video recordings from minimally invasive gynaecological surgeries. Endoscopic images were captured by using animal models at a clinically optimum illumination and focus with 720 × 576 pixels and 24 bits colour for (a) various testing targets from a colour palette with a known colour distribution, (b) different viewing angles, and (c) two different distances from calf endometrium by hysteroscopy and from poultry abdomen by laparoscopy. Human hysteroscopic and laparoscopic pelvic images from the endometrium and ovaries, respectively, were also captured and analysed. For texture feature analysis, three different sets were considered: (i) statistical features (SF), (ii) Spatial Gray Level Dependence Matrices (SGLDM), and (iii) Gray Level Difference Statistics (GLDS). All images were γ-corrected, and the extracted texture feature values were compared against the texture feature values extracted from the uncorrected images [1,10]. 

In another study, endometrium images in a hysteroscopy office from 40 women with 209 normal and 209 abnormal ROI were compared. Neural network models were also trained to classify 100 normal and 100 abnormal endometrial images with increased CAD sensitivity and specificity, however, without significant difference [1,10]. On the basis of the above results, a standardized protocol was proposed for image capture conditions to optimize tissue texture analysis. In a similar study, data of 52 patients were examined, and 516 ROIs were captured. The ROIs were equally distributed among cases of normal and endometrial hyperplasia. RGB images first γ-corrected and then converted into HSV and YCrCb reached 81% correct classification of ROI by using SF and GLDS features with an SVM classifier. 

Neophytou M et al., 2015, evaluated 52 hysteroscopic images of 258 normal and 258 abnormal ROIs extracted manually by the gynaecologist, and tissue diagnosis was verified by histopathology after biopsy [1]. The YCrCb colour system with SF+GLDS colour texture features based on SVM modelling could correctly classify 81% of the cases with a sensitivity and specificity of 78% and 81%, respectively, for normal and hyperplastic endometrium [1]. 

The imaging processes of four studies evaluating normal and abnormal tissue for the endometrium and cervix are summarized in Table 4. The third group of studies deals with hysteroscopic images and lacks the volume of data to extract concrete results. The review study of cervical cancer screening suggests an interesting algorithm, diagnosing the lesion margins by using a colour features discrimination process [13]. 

An analysis of hysteroscopic images of 28 patients with abnormal uterine bleeding and images of 39 patients without any pathologic features extracted 167 texture and vessel features for each image. Using artificial neural networks, four tissue features were selected to classify the images further. The specific software system verified the histopathologic diagnosis of 39 patients with normal endometrium and 10 patients with carcinoma with classification accuracy of 91.2%, and specificity and sensitivity of 83.8% and 93.6%, respectively [20]. 

## 4. Discussion

The camera systems, monitors, operative techniques, and skills developed with minimally invasive surgery provide tissue images and magnification with exceptional clarity. The abdomen and individual organs can be examined in situ with ease, without disturbing the anatomic features or the pathologic condition before treatment. In addition, video images can be used intra- and postoperatively to re-evaluate the pathologic condition and operative technique and for teaching purposes. They provide the surgeon with excellent quality real-time video, assessing cavities and areas of the human body impossible to observe with the naked eye. The easy access to tissue images facilitates, encourage, and accelerate the application of bioinformatics using different algorithms, which are correlated with the histopathological findings [1,10]. 

Dual-working channel endoscopes can enable an image-guided punch biopsy by using OCT. Matched OCT images obtained in vivo corresponding to histological biopsies can improve the accuracy and reliability of the technique [21]. An improvement in image resolution and the development of more specific imaging technology, such as polarization sensitive OCT, may also improve the accuracy of detecting buried pathologic features [21]. However, dual-working channel endoscopes increase the tip diameter of the scope, which is a big disadvantage when small cavities are observed, as in hysteroscopy. OCT is frequently used in ophthalmology and can provide information about cell architecture and morphology up to 15 nano microns below epithelial cells [14]. 

Tissue visual signs, image texture analysis, and selected features by electronic neural network systems can serve as biomarkers distinguishing abnormal from normal tissue. Precancerous as well as cancerous conditions are characterized as images with a complex set of attributes. Colour, texture, and relative geometry are predominately useful, while region shape is significantly less so. Regions are frequently amorphous, or, for a few region classes, exhibit a shape which may be only approximately modelled, and even in these cases, the model may be image dependent. The overall region of interest in the images may in general correlate with the histopathologic cancerous characteristics, such as abnormal tissue architecture, neo-angiogenesis, oedema, and cellular dysfunction. Images from a histopathologic section produced by microscopy may be interpreted by visual signs and tissue image features by computer-assisted diagnosis [22]. Such translation from microscopy tissue section characteristics to tissue image textures demand an allocation of data and computer system training [23]. CAD may have the potential to diagnose early disease, including cancer [1]. The loading of data with digital features of normal and abnormal tissue, with both visual and histopathologic characteristics, is essential in building the primary level of bioinformatics. The functionality and efficiency of CAD depends on network capacity, speed of data processing, and technological support [1]. 

The texture discrimination of capsule endoscopy (CE) video frames can be improved by modelling classical texture descriptors in the colour scale plane instead of the colour plane, as usually assumed by classical approaches [4]. Higher order statistics applied to the joint distribution of classical texture descriptors appear effective for texture characterization. Future work will include introducing different classification schemes [4]; augmenting the database, which is important in generalizing the results, especially when higher order statistics’ modelling is involved; exploring the temporal dynamics of texture information, since taking information from neighbour frames may improve classification performance [4]. 

Optical coherence tomography (OCT) is widely considered a real-time intraoperative tumour margin assessment because of its high-resolution (HR) images, rapid scanning, and optical properties [24]. However, although OCT provides HR images, the combination of OCTSS and spectral domain (SD) is still insufficient to effectively classify different types of internal organs [24]. The main reason is that OCT images are simply composed of the reflectivity of light (elastic scattering property), which can only reflect the texture information instead of molecular information [25]. OCT is a minimally invasive method to evaluate buried glands or other subsurface features and may be used to evaluate the efficacy of other endoscopic therapies, such as cryoablation and photodynamic therapies, not only in the GIT but also in the skin and abdominal cavity [26]. 

Raman spectrum is aimed at improving the accuracy of tissue margins’ delineation by detecting the margin of tumour surrounded by normal tissues, e.g., muscle. Based on the integrated system, OCT and RS can acquire the measurement with similar experimental conditions [27]. This allows for real-time review and assessment of the margins. Tumour margin detection can be evaluated with different algorithms and tissue types. 3D optical coaxial tomography and Raman spectroscopy were the two additional modalities used in combination with the tissue texture analysis to augment CATIA diagnostic ability. Coaxial tomography seems to provide extra information regarding the tissue cell layers below the superficial layer and can be used as an added tool to the optic system. Raman spectroscopy provides highly specific 3D spectra with intensity and time axes mainly used during microscopy for histopathologic sections. 

Although many ENT articles have been published on CAD, research on tissue texture analysis was missing. No studies using CAD for endometriosis were found. The intensity, density, and variety of tissue hue found in cases of pelvic and abdominal endometriosis would facilitate CATIA research in clinical practice. CATIA could probably contribute to the identification and quantification of endometriosis, especially the depth and extent of the disease on one tissue, the epithelium, and could probably assist in surgical treatment and the depth of destruction by laser and other modalities. Prospective and randomized studies are needed before CATIA is implemented in clinical practice. 

We found no prospective randomized studies published on CATIA. Most studies we reviewed failed to provide convincing evidence regarding the efficiency and efficacy of their image processing to distinguish normal from abnormal tissue, detect with high accuracy malignant tissues, and verify histopathologic results. The number of cases and image samples analysed in these studies were small, and the methods used were not well described. A major limitation of some studies was the absence of co-registered histologic features of lesions in system datasets, which was due to procedural difficulties; for example, using OCT to guide a biopsy during OCT imaging and using biopsy forceps with a small size and area coverage [4,11]. To address this limitation, one study compared the in vivo and ex vivo images of one biopsy from an endoscopic mucosal resection specimen obtained at the gastroesophageal junction [28]. Another limitation in cross sectional studies was comparing patients who were at different time points in treatment and stages of the disease. Variations in disease severity and responses across the patient population contribute to variations in data. In all studies, illumination was adjusted for optimal viewing but not for calibrating results to include the viewing angle, distance, and magnification of images. Experiments during hysteroscopy demonstrated that when three different texture feature algorithms, SGDLM, GLDS, and SF, were used, CATIA results were reliable when the distance of the telescope tip to the tissue target was within 3 cm and the viewing angle was kept within 15 degrees deviation [10]. 

Selecting the best algorithm or combination of algorithms for the diagnosis of malignant tissue and new cases was a major challenge in almost all studies. CATIA technology needs to be adapted to clinical use, with real-time image analysis supported by a physician-friendly interface. Use of this technology for the diagnosis of malignancy is to diminish false negative results, a fact that is usually accompanied by an increase in false positives and a reduction in specificity.

In the studies we reviewed, the major advantage of CATIA was comparing an abnormal tissue region to adjacent normal healthy tissue. However, the comparison between the healthy and the adjacent unhealthy area was neglected in the examination. Comparing images can be used during the intra- and postoperative period to re-evaluate the pathologic features and operative technique. Easy access to tissue images facilitates, encourages, and accelerates the application of bioinformatics by using different algorithms correlated with histopathologic findings [1,10].

## 5. Conclusions

CATIA results are encouraging, as many studies demonstrate the CAD systems’ potential to confirm, with high accuracy, abnormal tissue findings diagnosed by histopathology. Our review of CATIA research shows that much information can be extracted and used to diagnose and distinguish normal from abnormal tissue. The naked eye can analyse colour frequencies and detect shape and size differences of 100 microns in diameter. Minimally invasive surgery can facilitate and increase human sight limits up to 34 times that of the naked eye [28]. This is a great advantage to non-experienced eyes as well as in shortening the diagnostic time and starting treatment earlier.

CATIA and OCT will enable the evaluation of several cell layers now beneath human visual capacity. New technical and computational advances will improve optical biopsy and the precision of lesion excision during minimally invasive surgery. The exchange of knowledge, collaboration, identification of tasks, and CATIA method selection strategy will further improve CAD implementation in daily practice at a low cost. 

More extensive validation on a larger dataset, along with well-designed studies using CATIA, will be required once it is used in a clinical setting as a software system. When CATIA proves that it may increase the surgeon’s diagnostic ability and sampling precision, it could augment the intraoperative management decision and the surgeon’s performance. Additionally, it could minimize complications such as haemorrhage, haematoma, the spread of malignant cells, infection and scarring from multiple biopsies, as well as extensive tissue injuries. A proven efficacy of a CAD method discrimination ability, after validation by prospective and randomized studies, will allow the clinical implementation of CATIA systems and optical biopsies. 

## Figures and Tables

**Figure 1 jcm-10-05770-f001:**
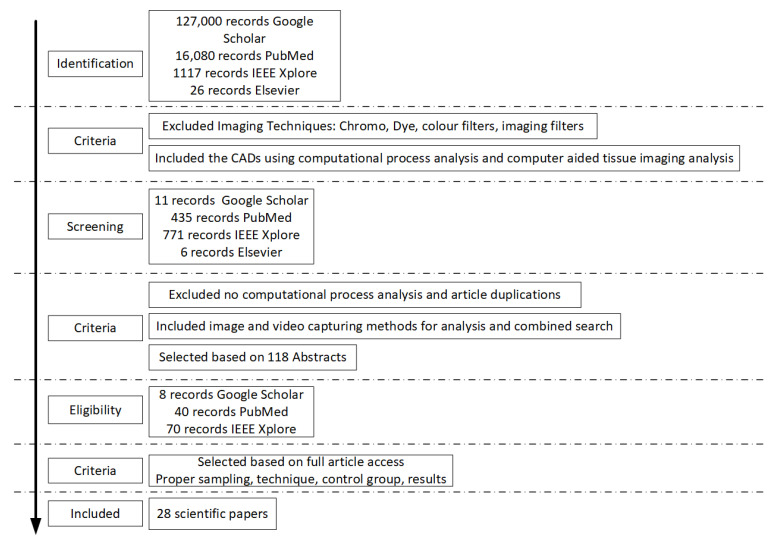
Summary of the search result papers and the evaluation procedure. Flow chart of the search strategy for the CATIA articles that appear in the literature.

**Table 1 jcm-10-05770-t001:** Summary and Indexing of tissue image processing terminology and techniques.

Image Processing Evaluation: Manual or automated image interpretation that filters artefacts from a database of images, e.g., endoscopic surgery video summary.
Colour-spectrum, Characterization and Filtering: Image colour texture content of the region of interest (ROI). Colour texture features are extracted over different colour spaces or hue saturation values (HSV). (1) Red green blue (RGB), (2) Luminance (Y), (3) Chrominance (red-yellow)/chrominance (blue-yellow) (YCrCb). For each colour space component, a standard grayscale feature is used and can be widely applied for texture characterization according to different texture features.
Tissue image features There are 26 texture features from each colour component. (i) *Statistical Features (SF):* (1) Mean, (2) Variance, (3) Median, (4) Energy, (5) Skewness, (6) Kurtosis, (7) Mode, (8) Entropy. (ii) *Spatial Gray Level Dependence Matrices (SGLDM)* (1) Angular second moment, (2) Contrast, (3) Correlation, (4) Homogeneity, (5) Variance, (6) Entropy, (7) Sum Entropy, (8) Sum Average, (9) Sum Variance, (10) Difference Entropy, (11) Difference Variance, (12) Information Measurement of Correlation 1, (13) Information Measurement of Correlation 2. (iii) *Gray Level Difference Statistics (GLDS)*: (1) Mean, (2) Entropy, (3) Contrast, (4) Homogeneity, (5) Energy.
Algorithm and Statistical evaluation Training and testing to distinguish normal from abnormal Regions of Interest (ROI). CATIA system performance was evaluated using *SVM algorithm* and *probabilistic neural networks (PNN)*. *C-SVM network* was used to investigate the Gaussian radial basis function (RBF) kernel and the linear kernel. *Principal component analysis (PCA)* reforms a dataset into a bilinear model of linear independent variables and uses a mathematical equation to explain the variation within the dataset. Vectors within the matrix are reshaped into images that show the spatial distribution forming abundance images, which represent the abundance of each vector for each pixel. Abundance images are then plotted in a colour-scaled image and can be combined with prominent differences between the samples highlighted. *Partial least squares discriminant analysis (PLS-DA)* is a supervised data reduction technique. It uses a versatile algorithm that can predict and describe modelling as well as select discriminative variables.

**Table 2 jcm-10-05770-t002:** Imaging process evaluating normal and abnormal tissue for dermatological pathologies.

Journal	Technique	Aim	Sample	Methodology	Results	Critical comments
Intl J of Scient & Engineering Research (IJSER) [5]	CATTA of microscopic images	Skin Cancer Dg (Squamous and Basal cell Carcinomas)	80 images of skin BMP 24 bit/pixel 1280 × 1024 pixels Blocks Size = 75–225 pixels	Compare normal and abnormal tissue (1) Pre-processing: RGB 2 GRAY - liner image stretching (2) Image ROIs partitioning (3) Discriminating texture features Fifty samples used for training the minimum distance classifier method	Combinations of textural types of features 200 × 200 pixels recognition provided 89.92% for allocation accuracy 93.75% for diagnosis accuracy	Small number of samples Image Stretching = loss of imaging info Partitioning = increase processing time and computational resources Image segmentation may increase the diagnostic accuracy
Photomed Laser Surgery, 2012 [6]	Raman spectroscopy and PCA algorithm	Discrimination of BCC and melanoma from normal skin biopsies in vitro	Histo skin images: 145 spectra Normal 30 BCC 96 Melanoma 19	Raman spectroscopy coupled to a fiber optic Raman probe. PCA and Elucidation distance classify samples according to histopathology	Differentiate normal from BCC and melanoma with accuracy 92.4%, sensitivity 99.1%, and specificity 93.3%	Raman spectroscopy can discriminate colour band frequency and can possibly be combined with any CATIA method adding to the sensitivity of the test. However, the technical complexity and procedural approach seem to be the main obstacles to its implementation.
Lasers in Surgery and Medicine, 2015 [7]	Raman spectroscopy and multivariate statistics	Discrimination between Mg and Bg skin lesions prior to surgery	In vivo image results compared to histopathology 250 normal images and 14 Bg, 133 BCC 30 SCC, 57 AK	PCA/DA and PLS/DA based on Euclidean Quadratic space distances were used to discriminate between Bg and Mg tissues using RS	Non melanoma versus normal and Bg precancerous lesions; the discrimination accuracy was 91.9% and 82.8%, respectively	Standard methodology microscopy Histopathological sections analysis
Series in Bio Engineering [8]	Illumination correction and feature extraction on skin lesion images	Skin lesion analysis Melanoma and nevi	Overall 206 standard camera images ROI 200 × 200 pixels 119 malignant melanomas and 87 nevi	(1) Multistage illumination correction algorithm (2) Histogram equalization (3) Feature extraction method (4) SVM model	Accuracy 72.52–81.26% for 3 different feature sets Combination between the features and framework provide better results	Data set is fair Conclusions are confusing Weak methodology (ground truth values, algorithms for nevi versus melanoma are not acknowledged) Insufficient data analysis Discrimination between malignant melanoma and nevi is not clear

Index: CATTA = Computer-aided tissue texture analysis, PCA = Principal component analysis algorithm, DA = discriminant analysis, PLS = Partial least squares, BCC = Basal cell carcinoma, ROIs = region of interest, BMP = Bitmap format, AK = actinic keratosis, Bg = benign, Histo = histopathology.

**Table 3 jcm-10-05770-t003:** CATA used in endoscopic procedures for the Gastrointestinal tract.

Journal	Targeted Organ	Technique	Aim	Sample	Methodology	Results	Critical Comments
Alinent Pharmacol Ther [2]	Colon	Colonoscopy vs. chromo endoscopy	Dg of dysplasia in patients with IBD	6 studies 1277 patients with IBD (Review paper)	Comparison of std colo/py to Chromo endoscopy to detect dysplasia controlled by histo Meta-analysis	91.3% specificity, 93.9% sensitivity for tumor detection	Not reported 1. Tu size detection ability 2. Texture features values 3. Small no of frames
World Journal of Gastrology [9]	Celiac Disease Diagnosis	Endoscopy and Computer Aided Texture Analysis (CATTA)	Detection of Intestinal mucosa alterations due to celiac disease	290 children 2835 duodenum	Endoscopic images recorded tissue alteration by modified immersion technique compared to histopathology Bx	CATTA reduced Dg error up to 31% Dg accuracy improved by 10%	Small patient numbers Low statistical power analysis Weak study design
BioMedical Engineering OnLine [10]	Colon polyps and colitis	Chromo endoscopy	Detection Rate of polyp and ulcerative colitis Procedure time	75 patients 586 images	White light endoscopy (WLE) followed by CE (Indigo Carmine) colonoscopy for UC surveillance	×30 ability to detect metaplasia 72% in pre-CE-IM and 63% in post-CE-IM	CATTA improves diagnostic accuracy Well designed study
Gastrointest Endosc. [11]	Esophagus	3D optical coherence tomography for CE-IM	Detection of esophageal metaplasia	patients 18 pre-treatment 16 post-RFA tx	Identification of metaplasia before and after therapy	Chromo endoscopy is superior to light endoscopy by 7% detecting dysplastic lesions	Heterogeneous samples More studies needed
BioMedEng OnLine [12]	Small bowel	Endoscopic capsule video multiscale wavelet	Detection of small Bg or Mg bowel tumors	14 patients 700 frames	Multiscale texture features analysis Wavelet transformation Image Classification	Dysplasia detected by WLE at 9.3% and WLE and CE at 21.3%. Improved Median colo/copy withdrawal time	high rates of polyp detection enhanced dysplasia detection

Index: Dg = Diagnosis, IBD = Inflammatory bowel disease, Bg = benign, Mg = malignant, Chromo endoscopy = Methylene Blue or Indigo Carmine, RFA tx = radiofrequency ablation treatment. Std = standard, Histo = histopathology, colo/py = colonoscopy, Bx = biopsies, Tu = tumor, CE-IM = Complete Eradication of Intestinal Metaplasia.

**Table 4 jcm-10-05770-t004:** Technical papers for image processing techniques.

Journal	Target Organ	Technique	Aim	Sample	Methodology	Results	Critical Comments
BioMedical Engineering OnLine, 2007 [10]	A standardized protocol for texture feature analysis of endoscopic images in gynecological cancer	Endoscopy video	Gynecology	Normal 209 vs. 209 abnormal ROIs	Texture Features Colour correction Gamma correction Calibration Viewing conditions	Gamma correction improve the comparison between different viewing conditions. Texture features can differentiate normal vs. abnormal ROIs	More images can be imported for further analysis
CBMS, 19th IEEE International Symposium on IEEE, 2006 [13]	Technology for medical education, Research, and Disease Screening by Exploitation of Biomarkers in a Large Collection of Uterine Cervix Images	Cervical images colour features discrimination	Gynecology		Classification using Gaussian mixture model, Lab colour and one geometrical feature to discriminate clinically significant images.	Image pre-processor used to remove specular reflecting artifacts with 90% success rates	New camera projection software with an algorithm to infer the rotation of the lens improved boundary estimation and image conversions
IEEE EMBS 2009 [14]	Texture-based Computer-Assisted Diagnosis for fiberscope Images	GIT endoscopy Fiberscope images	For Improving diagnosis in GIT Endoscopy Images before and after CATA	350 esophagus 129 gastric mucosa 164 Barret esophagus 158 squamous epithelium	A new CAD system that filters the artifacts first with an image filtering algorithm, then applies a colour texture algorithm. Evaluation is based on an image database with artificially rendered fiber artifacts	Similar to highest accuracy achieved by standard original images and Gabor filter by 80% No improvement after filtered procedure	Insufficient pre-processing More tissue texture feature algorithms can be applied
IEEE International Conference on Image Processing, 2006 [15]	Hierarchical Summarization of Diagnostic Hysteroscopy Videos	Hysteroscopy video	Gynecology	12 hysteroscopy videos	Video summarization Video segmentation	False positive 26% for 11 videos	Small sample More videos
IEEE EMBS, 2010 [16]	An Integrated Port Camera and Display System for Laparoscopy	Port camera	Gynecology		The Powered port camera integrating visual system components with a cannula port	Results show that ex vivo tissue identification and acquisition was as good as the traditional methods	New device: (1) reduced the invasiveness of the laparoscopic procedure (2) reduced its cost (3) improved the laparoscopic procedure

## Data Availability

Not applicable.

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
