# Peer review of "Is Computer-Assisted Tissue Image Analysis the Future in Minimally Invasive Surgery? A Review on the Current Status of Its Applications"

_jcm, 2021, doi:10.3390/jcm10245770_

Round 1

Reviewer 1 Report

It is clearly a work showing us the future, when the AI will process the image and reach a real "optical biopsy", but it is still early. Dealing with oncology there is no place for false negative results. 

There is another optic technology "Narrow Band Image", developed 20 years ago and validated for early endoscopy detection of gastric cancer. There also studies in uro-epitelium and endometrium. Personally have experience with this technology and kappa index where compared with histology was high. I added some references in case you would like to use it.

Personally I believe that it will be of a limited use during diagnosis by direct vision, only for helping non experienced eyes or to shorten the time to diagnosis and starting treatment earlier. Biopsies are always taken if the image is not of normality. 

The most interesting application is as a helping tool during surgery giving information about the optimal surgery by reviewing in real time the margins and be sure that margins are free of disease.

Congratulation for bringing this review of the potential use of the technology and creating expectation about what is coming to help the endoscopists. 

Finally marked on the pdf some minor english correction that are still needed.

Horiuchi Y, Tokai Y, Yamamoto N, Yoshimizu S, Ishiyama A, Yoshio T, Hirasawa T, Yamamoto Y, Nagahama M, Takahashi H, Tsuchida T, Fujisaki J. Additive Effect of Magnifying Endoscopy with Narrow-Band Imaging for Diagnosing Mixed-Type Early Gastric Cancers. Dig Dis Sci. 2020 Feb;65(2):591-599. doi: 10.1007/s10620-019-05762-9. Epub 2019 Jul 31. PMID: 31367881.

Kisu I, Banno K, Kobayashi Y, Ono A, Masuda K, Ueki A, Nomura H, Hirasawa A, Abe T, Kouyama K, Susumu N, Aoki D. Narrow band imaging hysteroscopy: a comparative study using randomized video images. Int J Oncol. 2011 Nov;39(5):1057-62. doi: 10.3892/ijo.2011.1131. Epub 2011 Jul 18. PMID: 21769430.

Kisu I, Banno K, Kobayashi Y, Ono A, Masuda K, Ueki A, Nomura H, Hirasawa A, Abe T, Kouyama K, Susumu N, Aoki D. Flexible hysteroscopy with narrow band imaging (NBI) for endoscopic diagnosis of malignant endometrial lesions. Int J Oncol. 2011 Mar;38(3):613-8. doi: 10.3892/ijo.2011.903. Epub 2011 Jan 14. PMID: 21240458.

Author Response

Responses to comments made by Reviewer #1

  1. It is clearly a work showing us the future, when the AI will process the image and reach a real "optical biopsy", but it is still early. Dealing with oncology there is no place for false negative results.

Response: We would like to thank Reviewer #1 for reviewing our paper and recognizing the benefits of our work upon the existing literature. We agree with this statement and add that it is especially hard in cases which are balanced at the early stage. We assure the reviewer that we have considered all his comments and elaborate further and in detail on our changes.

  1. There is another optic technology "Narrow Band Image", developed 20 years ago and validated for early endoscopy detection of gastric cancer. There also studies in uro-epitelium and endometrium. Personally, have experience with this technology and kappa index where compared with histology was high. I added some references in case you would like to use it.
  2. Horiuchi Y, Tokai Y, Yamamoto N, Yoshimizu S, Ishiyama A, Yoshio T, Hirasawa T, Yamamoto Y, Nagahama M, Takahashi H, Tsuchida T, Fujisaki J. Additive Effect of Magnifying Endoscopy with Narrow-Band Imaging for Diagnosing Mixed-Type Early Gastric Cancers. Dig Dis Sci. 2020 Feb;65(2):591-599. doi: 10.1007/s10620-019-05762-9. Epub 2019 Jul 31. PMID: 31367881.
  3. Kisu I, Banno K, Kobayashi Y, Ono A, Masuda K, Ueki A, Nomura H, Hirasawa A, Abe T, Kouyama K, Susumu N, Aoki D. Narrow band imaging hysteroscopy: a comparative study using randomized video images. Int J Oncol. 2011 Nov;39(5):1057-62. doi: 10.3892/ijo.2011.1131. Epub 2011 Jul 18. PMID: 21769430.
  4. Kisu I, Banno K, Kobayashi Y, Ono A, Masuda K, Ueki A, Nomura H, Hirasawa A, Abe T, Kouyama K, Susumu N, Aoki D. Flexible hysteroscopy with narrow band imaging (NBI) for endoscopic diagnosis of malignant endometrial lesions. Int J Oncol. 2011 Mar;38(3):613-8. doi: 10.3892/ijo.2011.903. Epub 2011 Jan 14. PMID: 21240458.

Response: We thank Reviewer for the comments and recommendations. This is a very interesting field of research as narrow band image (NBI) can enhance and magnify visualisation and therefore assist the surgeon significantly. However, after reviewing the recommended references, we reply with modesty that papers using NBI analysis were excluded from our study since NBI does not fulfil the criteria of computer assisted tissue image analysis which is a computation analysis.

  1. Personally I believe that it will be of a limited use during diagnosis by direct vision, only for helping non experienced eyes or to shorten the time to diagnosis and starting treatment earlier. Biopsies are always taken if the image is not of normality. 

Response: We completely agree with Reviewer. The use of an image analysis algorithm will be supported for the physician to increase his/her accuracy during the investigation not to substitute the surgeon.

See Page 12, Section 5, Conclusions, line 560

“This is a great advantage to the non-experienced eyes as well as in shortening the diagnostic time and starting treatment earlier.”

See Page 12, Section 5, Conclusions, line 568

“When CATIA proves that it may increase the surgeon’s diagnostic ability and sampling precision, it could augment the intraoperative management decision and the surgeon’s performance.”

  1. The most interesting application is as a helping tool during surgery giving information about the optimal surgery by reviewing in real time the margins and be sure that margins are free of disease.

Response: We completely agree with Reviewer. This is incorporated in our pre-existing explanation on margin evaluation by Raman Spectrosocpy. In this paper we have included the reference [23] because in this specific article CATIA and Raman methods have been used in combination. 

Reference [23] - [Chih-Hao Liu, Ji Qi, Jing Lu, Shang Wang, Chen Wu, Wei-Chuan Shih, Kirill V. Larin, “Improvement of tissue analysis and classification using optical coherence tomography combined with Raman spectroscopy”, Journal of Innovative Optical Health Sciences, Vol. 08, No. 04, 2015.] 

            See Page 11, Section 4, Discussion, line 462

“Raman spectrum aimed to improve the accuracy of tissue margins delineation by detecting the margin of tumour surrounded by normal tissues, e.g., muscle. Based on the integrated system, OCT and RS can acquire the measurement with more similar experimental conditions [23]. This allows for real time review and assessment of the margins.”

  1. Congratulation for bringing this review of the potential use of the technology and creating expectation about what is coming to help the endoscopists. 

Response: We thank Reviewer for appreciating our work and effort.

  1. Finally marked on the pdf some minor english correction that are still needed.

Response: We thank Reviewer for addressing these minor mistakes. We have updated as requested. The changes were added in the paper with track changes for easy review.

Reviewer 2 Report

This was quite well written review. However I detect a few issues that I would like you to change or explain them.

First of all the title. It neither indicates any problems to be discussed, nor raises any questions. It only states " Computer-assisted Analysis of Tissue Images", I find it not enough to encourage future readers.

Seconds, the tables: in first collumn there is often a statement "Error! Reference not found" - it should not been published so.

M& M section - you clearly described how you choose the articles but I felt a little bit confused - what for was the whole review? to describe the CATIA? to discuss statistical issues? why three so distant specializations were choosen? 

Author Response

Responses to comments made by Reviewer #2

  1. This was quite well written review. However, I detect a few issues that I would like you to change or explain them.

Response: We would like to thank Reviewer for reviewing our paper and recognizing the benefits of our work upon the existing literature. We assure that we have considered all his comments and elaborate further and in detail on our changes

  1. First of all the title. It neither indicates any problems to be discussed, nor raises any questions. It only states " Computer-assisted Analysis of Tissue Images", I find it not enough to encourage future readers.

Response: We thank Reviewer for highlighting this point. We appreciate what they mean by it and have therefore modified the title. The title was rephrased so as to be more relative to the methodology and catchier.

Page1, Title, Line 1

“Is Computer-Assisted Tissue Image Analysis the future in minimally invasive surgery? A review on the current status of its applications.”

  1. Seconds, the tables: in first column there is often a statement "Error! Reference not found" - it should not be published so.

Response: We thank Reviewer for highlighting this point. The changes were added in the paper with track changes for easy review.

  1. M & M section - you clearly described how you choose the articles, but I felt a little bit confused - what for was the whole review? to describe the CATIA? to discuss statistical issues? why three so distant specializations were chosen? 

Response: We thank Reviewer for highlighting that this part of the manuscript might be confusing. We added in the introduction section the following paragraph to support and clarify the purpose of our study.

The purpose of the study was to provide a review of computer-assisted tissue image analysis studies during minimally invasive surgery and endoscopy. In addition, we review and evaluate the impact of in vivo optical biopsies performed by tissue image analysis on the surgeon’s diagnostic ability and sampling precision.

see Page 3, Section 1, Line 85

Following the Material and Methods Section 7, Page 3, line 97 is written “Our search further specifically isolated the articles in the disciplines of dermatology, gastroenterology, otorhinolaryngology (ENT), and gynaecology.” we could not find any more relevant to CATIA articles than those reported and analyzed in this paper. The research on CATIA was very limited hence all medical disciplines used CATIA were included in our evaluation in order to extract safe conclusions.   

Round 2

Reviewer 2 Report

thank you for all implemented changes

This manuscript is a resubmission of an earlier submission. The following is a list of the peer review reports and author responses from that submission.